# Rapid Detection of Tools of Railway Works in the Full Time Domain

Zhaohui Zheng [1,2] , Yuncheng Luo [1,*], Shaoyi Li [1,2], Zhaoyong Fan [1,2], Xi Li [1,2], Jianping Ju [1,2], Mingyu Lin [3] and Zijian Wang [1]

1   School of Information and Artificial Intelligence, Nanchang Institute of Science & Technology, Nanchang 330108, China
2   Nanchang Key Laboratory of Information Visualization Technology of Internet of Things, Nanchang Institute of Science & Technology, Nanchang 330108, China
3   School of Artificial Intelligence, Hubei Business College, Wuhan 430070, China
*   Correspondence: lam9130a@gmail.com

**Abstract:** Construction tool detection is an important link in the operation and maintenance management of professional facilities in public works. Due to the large number and types of construction equipment and the complex and changeable construction environment, manual checking and inventory are still required. It is very challenging to count the variety of tools in a full-time environment automatically. To solve this problem, this paper aims to develop a full-time domain target detection system based on a deep learning network for difficult, complex railway environment image recognition. First, for the different time domain images, the image enhancement network with brightness channel decision is used to set different processing weights according to the images in different time domains to ensure the robustness of image enhancement in the entire time domain. Then, in view of the collected complex environment and the overlapping placement of the construction tools, a lightweight attention module is added on the basis of YOLOX, which makes the detection more purposeful, and the features cover more parts of the object to be recognized to improve the model. Overall detection performance. At the same time, the CIOU loss function is used to consider the distance fully, overlap rate, and penalty between the two detection frames, which is reflected in the final detection results, which can bring more stable target frame regression and further improve the recognition accuracy of the model. Experiments on the railway engineering dataset show that our RYOLO achieves a mAP of 77.26% for multiple tools and a count frame rate of 32.25FPS. Compared with YOLOX, mAP increased by 3.16%, especially the AP of woven bags with a high overlap rate increased from 0.15 to 0.57. Therefore, the target detection system proposed in this paper has better environmental adaptability and higher detection accuracy in complex railway environments, which is of great significance to the development of railway engineering intelligence.

**Keywords:** railway instrument detection; deep learning; object detection; luminance enhancement

## 1. Introduction

Works departments are the foundation of the entire railway transport system. With the continued development of railway construction, the numbers and types of tools and materials used in the maintenance of railway equipment are also greatly increased, which makes the equipment maintenance of the rail system and its maintenance management more difficult. The traditional methods of railway on-site tool inventory are carried out by artificial inspection, which is complicated with the greater numbers and types of tool materials, especially in high-speed railways. Equipment maintenance work often occurs at night. Due to the lack of open vision, the affected sight and the uneven quality of personnel, it is easy to cause detection errors or even loss, even forming an important unfavorable factor that endangers railway safety. To date, the traditional management of the tools and materials has mainly adopted manual management. There is no uniform management

mode and standard, and there is no intelligent system management, especially in recent years. With the rapid development of high-speed railways, if the items are left on the site, they will cause serious harm to the train and equipment on site and even cause a serious traffic accident. In order to enhance the safety of railways, the authors have previously focused on detecting obstacles in railway tracks [1–3] and detecting rail integrity [4,5] but did not improve the problem of missing tools in the operation and maintenance link. The public works department is the foundation of the entire railway transportation. We must solve the possible hidden dangers from the operation and maintenance link. Machine learning methods can provide automatic inventory tools to ensure the security of operation and maintenance. Among them, the detection methods based on deep learning have received great attention. They are used in many application scenarios, such as pedestrian detection [6,7], face recognition [8,9], human pose estimation [10–12], etc., and they have achieved good application effects. However, in the environment of railway operation and maintenance, tool detection has its research difficulties for target detection algorithms: (1) with the continuous increase in railway construction and operation mileage, the types of tools and materials used in railway operation and the maintenance of railway signal equipment increase greatly; (2) the railway operation and maintenance occur at all times of the day, especially at night, and the instrument detection has higher adaptability requirements to object detection. (3) There is the overlap and occlusion problem of tool placement. To solve these problems, this paper has proposed a full time domain tool detection system-based deep learning network for difficult complex railway environment image recognition, whose main contributions include the following:

1.  An image enhancement network with brightness channel decisions is proposed to ensure the robustness of image enhancement in the full time domain.
2.  It has been proposed to add a lightweight attention module and use the improved CIoU loss function on the basis of YOLOX, which greatly improves the detection performance in the complex environment of tool collection and the overlay of tools.
3.  Compared with previous methods, this method has achieved encouraging experimental results on railway tool datasets in actual operation and maintenance scenarios.

## 2. Related Work

With the continuous development of object detection in deep learning, the technology is becoming increasingly mature. Neural networks have been applied to many fields in daily life, and railway operation and maintenance are already one of them. Object detection algorithms based on deep learning can be divided into two types: one-stage algorithms and two-stage algorithms. Among them, the classical two-stage object detection algorithms include R-CNN, Fast R-CNN, and Faster R-CNN [13–15], which are characterized by high accuracy and slow operation speed. A double-model rail inspection system (DM-RIS) is established for surface defects in [4], where FRGMM is presented for segmentation proposal and Faster R-CNN is utilized for objective location in a parallel structure, however, its detection target is relatively single, it is difficult to appear overlapping, occlusion and other phenomena between multiple targets. In comparison, the one-stage object detection algorithm runs quickly and conforms more to the practical application requirements. The representative algorithms include the SSD algorithm and YOLO series algorithm [16–21]. Wang et al. [5] has enhanced the performance of YOLOv2 by adjusting the detection layer of a single machine network so that it could inspect the key parts of railway tracks in real-time. Bai et al. [22] proposed an improved railway surface defect detection method based on YOLO. This method takes MobileNetv3 as the backbone to extract image features and applies depth-separable convolution to the Panet layer of YOLOv4 to realize a lightweight network and real-time detection of railway pavement. Chandran et al. [23] combines image processing and enhanced deep learning for the detection of railway fastening systems, proposing an image processing technique that can improve fastener position and remove redundant information from fastener images. Guo et al. [24] propose a real-time computer vision framework that improves YOLOv4 to have better accuracy and processing speed

than other models when inspecting missing or damaged track components. The above algorithms are collected by a collection vehicle with auxiliary lighting, and the data are relatively regular and orderly, but it is difficult to achieve excellent results for non-standard collection in a more complex environment. Guan et al. [1] proposed a lightweight three-stage network detection framework for railway obstacle detection. This method is composed of the CRP module, RODNET network and post-processing stage, and RODNET network constructs a much lighter and more effective detection model by improving YOLOv4-Tiny. He et al. [2] propose a YOLO-based rail obstacle detection model and embed it onboard to obtain an efficient test result. Chen et al. [3] proposed an efficient two-stage foreign object detection framework for railway images. In the first stage, a lightweight railway image classification network is established, and the input railway images are divided into two categories: normal railway images and intrusion railway images. To achieve accurate real-time classification, selective kernel convolution and convolutional block attention module (CBAM) are implemented on the inverse residual elements. Then, if the image is classified into the "hacked" class, it will be fed into the second-stage foreign object detection network to further detect the location and class of objects in the image. Lv et al. [25] improved the YOLOv5s deep learning framework for railway pedestrian detection. They added the L1 regularization function to the BN layer, which reduced the size of the model and improved the inference speed, and applied an attention module to correct the target position deviation in the process of feature extraction. The above algorithms use the reference object train track, and most of the data are in the daytime background, so they are not applicable to target recognition without reference objects at different times. There are relatively few studies on the object detection of tools in railway operation and maintenance. According to the research status of object detection, this paper adopted YOLOX as the fundamental object detection network and, on the original basis, designed a target detection system with high precision and high recognition speed according to the specific needs of railway operation and maintenance, which can meet these needs.

## 3. Proposed Method

Due to the full time domain of railway operation and maintenance and the large number of tools, the brightness difference of the railway tool dataset is large, and many superpositions are formed, which leads to the difficulty of object detection. To meet this challenge, we build the framework of railway tool recognition method in the full time domain by adaptively enhancing the dataset and improving the YOLOX detection module and loss function. As shown in Figure 1, we first increase the brightness enhancement network on YOLOX to enhance the images under a variety of different lighting conditions, then apply a lightweight attention module to reduce the influence of noise caused by image enhancement on network training, and finally modify the loss function to reduce the impact of the mutual superposition of objects.

### 3.1. Image Enhancement Module

The repair and maintenance of railway equipment usually needs to be carried out in the field at any time of the day. Therefore, there is a large difference in the brightness of the collected tool images, as shown in Figure 2. Because the brightness difference will have a great impact on the subsequent enhancement, we need to perform brightness enhancement operations on the original image. Common methods include adjusting the brightness and contrast through algorithms to enhance brightness, and the brightness enhancement neural network used in this paper. Raising the brightness by adjusting the brightness and contrast often brings about the problem of high noise. We add a brightness decision channel to RRDnet, divide the image into three different categories: dark image, medium brightness image, and bright image, and use different strategies to enhance the image, so this paper proposes dividing the images into three different categories, namely, low-brightness images, medium-brightness images and high-brightness images, and using different strategies to

enhance the images based on RRDnet [26]. We perform threshold discrimination by using the statistics to determine the brightness type.

$$T = (m_t - T_t)/T_t \tag{1}$$

where $m_t$ represents the image average brightness and $T_t$ is the global average brightness of the expected normal image. If $T < \tau t_1$ this image will be judged as a low-brightness image if $T > \tau t_2$ it will be judged as a high brightness image and if $\tau t_1 < T < \tau t_2$, it will be judged as a medium brightness image. $\tau t$ is the threshold used to determine the image brightness. This paper determined the most suitable values of $T_t = 0.45$, $\tau t_1 = 0.4$, and $\tau t_2 = 0.2$.

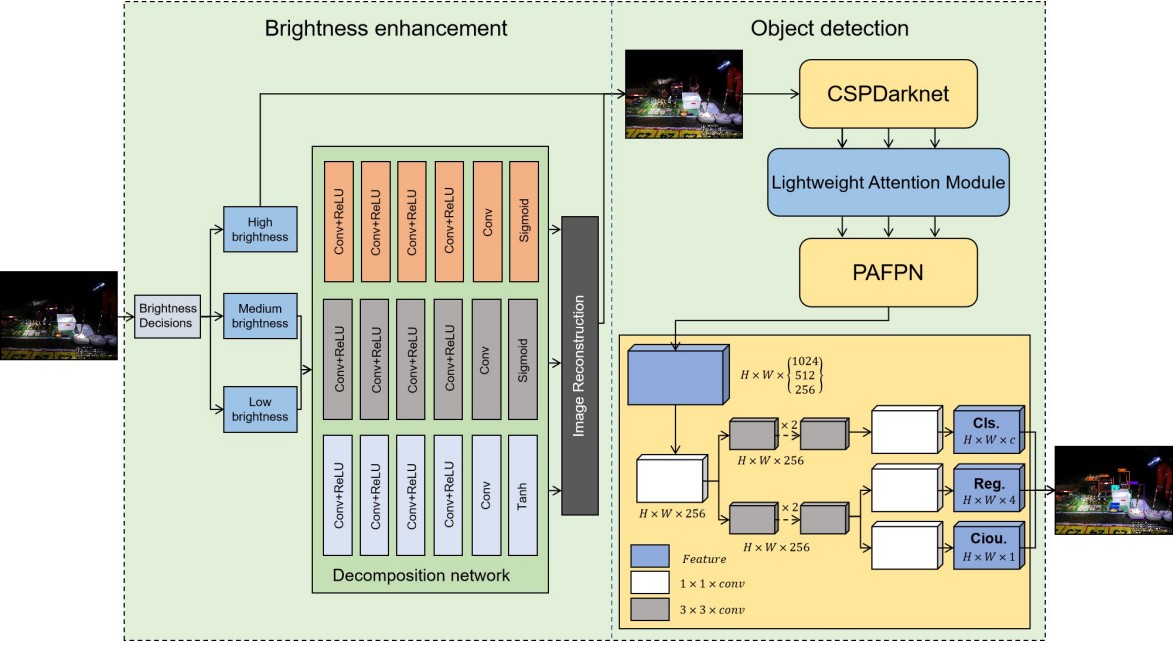

**Figure 1.** Our model framework.

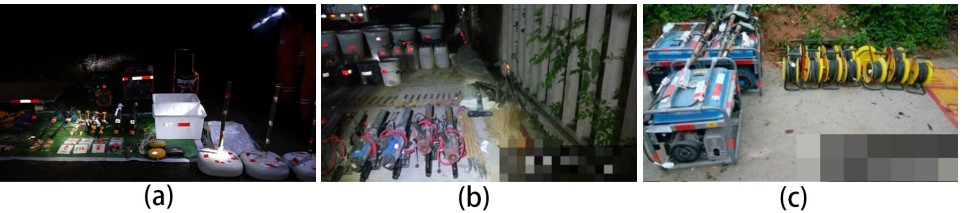

(a)      (b)      (c)

**Figure 2.** (**a**) The tool image of the low-brightness image. (**b**) The tool image of the medium-brightness image. (**c**) The tool image of the high-brightness image.

We add a brightness decision layer to the RRDnet brightness enhancement neural network for brightness enhancement and denoising. No processing is performed for high-brightness images, but we use different strategies for brightness enhancement and denoising for medium-brightness and low-brightness images. The network is a new type of three-branch convolutional neural network that decomposes the image into illumination components, reflection components and noise components. By designing an iterative loss function without image pairs, noise can be effectively estimated, and illumination can be recovered. The general idea is as follows: first, the unexposed image is decomposed into three branches through the decomposition network, namely, the estimated reflection component, estimated illumination component and noise component. Then, in the reconstructed network, the lighting components are processed through a gamma transformation. The reflection component is the result of the input image minus the noise component

divided by the illumination component. Finally, the image is reconstructed according to the Retinex [27] theory. For weight updates, a three-part loss function is used:

$$L = L_r + \lambda_t L_t + \lambda_n L_n \tag{2}$$

where $L_r$ is the reconstruction loss component based on Retinex, $L_t$ is the smoothing loss component of the illumination map, $L_n$ is the denoising loss component based on illumination map guidance, and $\lambda_t$ and $\lambda_n$ are the corresponding weight factors.

Since the low-brightness image and medium-brightness image have different requirements for the noise suppression, smoothing force and gamma transformation, the low-brightness image usually has more noise and weaker texture information than the medium-brightness image, and thus, we can better construct the appropriate loss function of the two types of images by setting different $\lambda_t$, $\lambda_n$ and gamma values, where $\lambda_t$ and $\lambda_n$ of the low-brightness images are larger than those of medium-brightness images, and the value of the gamma transformation is lower than that of medium-brightness images. In this paper, repeated experiments are designed: the extreme value of gamma is set to 0.1 to 1, and the step size is 0.1; the extreme value of $\lambda_t$ is set to 1 to 3, and the step size is 1; the extreme value of $\lambda_n$ is set to 3000 to 10,000. The results are analyzed to determine that the value of the gamma transformation of the dark image is 0.1, $\lambda_t = 1$, and $\lambda_n = 8000$, and those of medium-brightness images were determined as 0.4, $\lambda_t = 2$, and $\lambda_n = 5000$. The image enhancement effect is shown in Figure 3, which shows that the brightness enhancement of image samples can greatly improve the validity of the dataset.

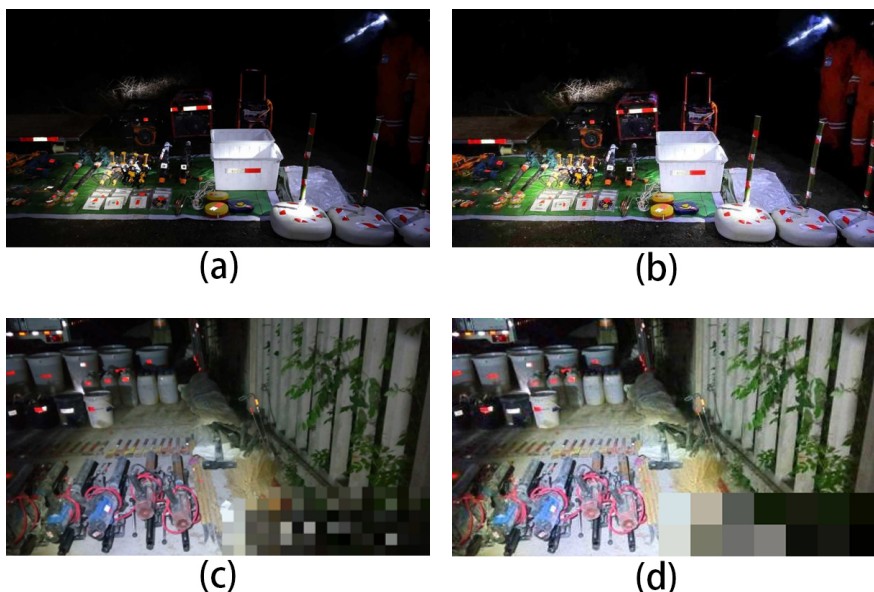

**Figure 3.** (**a**) The tool image of the low-brightness image. (**b**) The enhanced result of the low-brightness image. (**c**) The tool image of the medium-brightness image. (**d**) The enhanced result of the medium-brightness image.

### 3.2. Lightweight Attention Module

The cross stage partial (CSP) layer of YOLOX contains large residual error, and the residuals of operation could effectively avoid the deep web gradient disappearing and would together send the noise into the deep web; meanwhile there are a large number of the low-brightness images among our image samples, and some noise will also be introduced after image enhancement, which intensifies the influence of network training. Therefore, a lightweight attention module similar to the convolutional block attention module (CBAM) [28] is added to the CSP layer, which can reduce the influence of noise on network training by exerting attention and adjusting the weight of each channel.

For a given feature map, the lightweight attention module sequentially infers the attention map along the two independent dimensions of channel and space and then multiplies the attention map with the input feature map to perform adaptive feature refinement. For the specific operations below, channel attention processing is first carried out, and the input feature layer $X \in R^{H \times W \times C}$ is divided into G groups along the channel dimension. Then, the feature mapping in each channel is aggregated through the global average pooling operation (GAP) to generate the corresponding weight information. To adaptively generate the attention weight between channels and improve the computational efficiency, a multilayer perceptron (MLP) with a hidden layer and ReLU activation function are used, and the sigmoidal activation operation is also used after MLP processing. The calculation formula of the channel attention weight generated by each group is as follows:

$$W_{\text{chan}}(k) = \sigma(\delta(\text{MLP}(GAP(X_k)))) \quad k \in \{1, 2, 3, \cdots G\} \tag{3}$$

where $\delta$ is the ReLU function, $\sigma$ is the sigmoidal function, and $x_k$ is the input features for the first $k$ group.

Then, the input of each group adopts the residual structure to avoid the disappearance of deep feature values and performance degradation caused by deep layers. Therefore, the attention weighting formula for each channel group is as follows:

$$X'_k = W_{\text{chan}} \otimes X_k \oplus X_k \tag{4}$$

where $\otimes$ represents the multiplication of elements, and $\oplus$ represents the addition of elements.

The outputs from each group are connected into a new feature map $Y \in R^{H \times W \times C}$ as the input feature map of the spatial attention processing module, and we transform the feature map Y into the feature vector and then obtain the spatial information weight of $H * W * 1$ through the convolution and summation operation to prevent the deviation of sample data from being too large and affecting the network generalization; therefore, the weight of spatial information is normalized as follows:

$$W_{spat} = \sigma(\delta(f(Y))) \tag{5}$$

where $f$ is the convolution operation, $\delta$ is the ReLU function, and $\sigma$ is the sigmoidal function.

Finally, the reconstructed feature map and the input feature map are fused to obtain the output of the final attention module. The formula is as follows:

$$Y' = Y \otimes W_{spat} \oplus X \tag{6}$$

The lightweight attention module flow is shown in Figure 4.

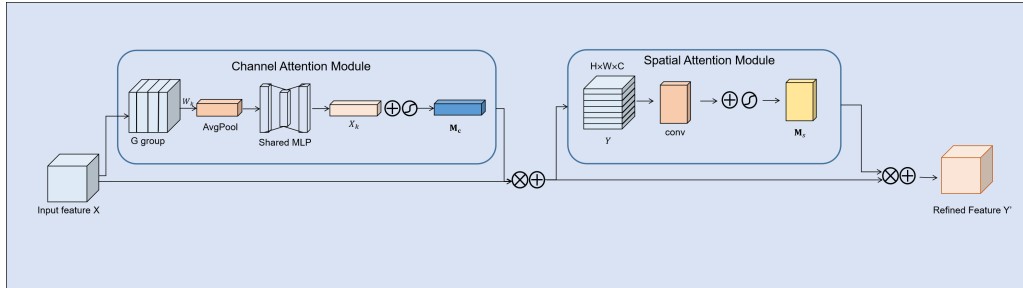

**Figure 4.** Lightweight attention module flow.

### 3.3. Loss Function

The loss function affects the convergence effect of the model and the fit effect of the evaluable algorithm. In view of the phenomenon of a large number of objects superimposed on each other in the tool images, as shown in Figure 5, the CIoU loss [29] function is used

in this paper to replace the positional loss function in the YOLOX network; in this manner, it can improve the model detection ability for densely distributed superimposed objects.

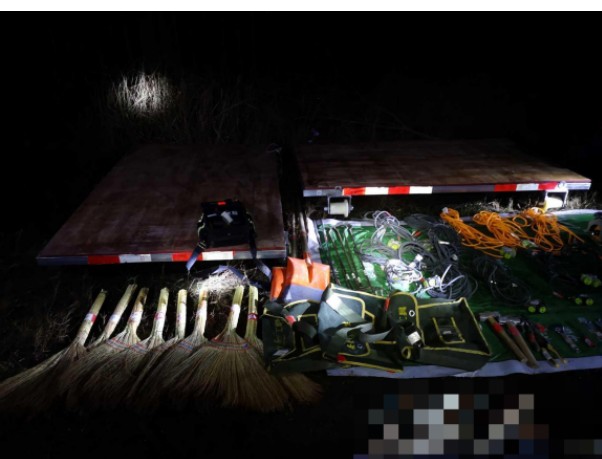

**Figure 5.** There are images of objects superimposed on each other.

The GIoU loss [30] algorithm has solved the problem that loss is equal to 0 when the boundary box does not intersect, but the convergence is slow in the horizontal and vertical directions when the two boxes intersect. The DIoU loss [29] algorithm takes the disadvantage of GIoU loss into consideration on the basis of the overlapping area features and directly optimizes the Euclidean distance of the centre points of the two boxes to accelerate convergence. The CIoU loss algorithm increases the size loss, length loss and width loss of the detection box on the basis of DIoU loss so that the prediction box can be more consistent with the real frame. The punishing items of CIoU loss are:

$$\mathcal{R}_{CIoU} = \frac{\rho^2\left(b, b^{gt}\right)}{c^2} + \alpha v \tag{7}$$

where $b$ and $b^{gt}$ represent the prediction box and the real box, $\rho^2$ is the square of the distance between the prediction box and the real box, $c^2$ is the square of the diagonal length of the smallest box that can just contain the prediction box and the real box, $v$ is used to measure the consistency of the relative proportion of the two rectangular boxes, and $\alpha$ is the weight function:

$$v = \frac{4}{\pi^2}\left(\arctan\frac{\omega^{gt}}{h^{gt}} - \arctan\frac{\omega}{h}\right)^2 \tag{8}$$

$$\alpha = \frac{v}{(1 - IoU) + v} \tag{9}$$

The full CIoU loss function is:

$$\mathcal{L}_{CIoU} = 1 - IoU + \frac{\rho^2\left(b, b^{gt}\right)}{c^2} + \alpha v \tag{10}$$

By improving the loss function, the model improves the accurate detection of stacked tools in images.

## 4. Experimental Results and Analysis

### 4.1. Preparation for the Experiment

The railway works datasets on the network are limited. This paper attains a homemade dataset of construction tools based on data obtained from a railway system, with a total of 351 images. The LabelImage tool is used to construct labels, including cart, generator, broom, electric drill, cable, water pipe, woven bag, toolkit, bucket, blower, tape measure, shovel, sand bucket, electric sander, plastic bucket, cement bucket. The 16 most commonly

used tools were selected as detection objects. Since the existing datasets are all from actual scenarios and the data scenarios are complex and diverse, there is no need to expand the datasets. We randomly divided the dataset into a training set and a test set at a ratio of 9:1 and then divided 10% of the training set into a validation set in the same manner.

The experimental environment is as follows: the operating system is an Ubuntu18.04.5LTS, the CPU is an Intel® Xeon® E5-2660 V3 @ 2.60 GHz, the video card is a GeForce GTX-1080Ti, the memory is 16 GB, The programming libraries used mainly include Pytorch1.8, numpy1.23.1, opencv_python4.6.0.66, matplotlib3.5.1, pillow9.2.0, tqdm4.64.0.

### 4.2. Evaluation Indicators

In this paper, the common evaluation indices of the target detection model, namely, average accuracy (mAP) and frames per second (FPS), are adopted as the evaluation indices of the tool detection model. The AP refers to the area under the precision-recall (PR) curve, and mAP refers to the mean value of the AP of each category. The larger AP and mAP values are, the better. The calculation formulas for precision and recall are as follows:

$$\text{Precision} = \frac{TP}{TP + FP} \tag{11}$$

$$\text{Recall} = \frac{TP}{TP + FN} \tag{12}$$

where *TP* (true positives) refers to those that were correctly identified as positive, *FP* (false-positives) refers to those incorrectly identified as positive, and *FN* (false negatives) indicates those that were incorrectly identified as negatives.

### 4.3. Analysis of Experimental Results

In the training process, the loss curve of our model is shown in Figure 6. It can be found from the figure that the loss curve gradually tends to be stable with the increasing number of training rounds. When the epoch reached approximately 80, the model gradually converged, and no overfitting phenomenon occurred in the training process. The confusion matrix (condidence = 0.25, iou = 0.65) of the model is shown in Figure 7.

To verify the tool detection performance of our model for the full time domain, we designed a set of ablation experiments and a set of comparison experiments. We tested the network performance effects of different modifications in this paper by an ablation experiment and then performed a comparison experiment between our model and mainstream networks (Retinanet, YOLOv5s, and YOLOX) to comprehensively analyse the performance of the model.

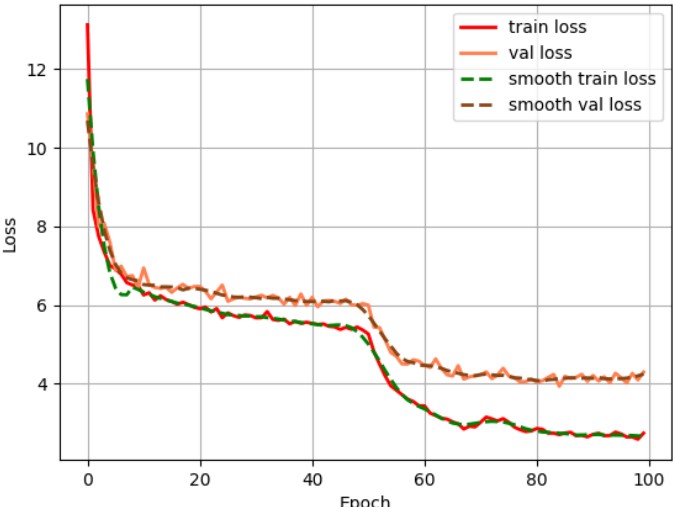

**Figure 6.** The loss value of the our model changes.

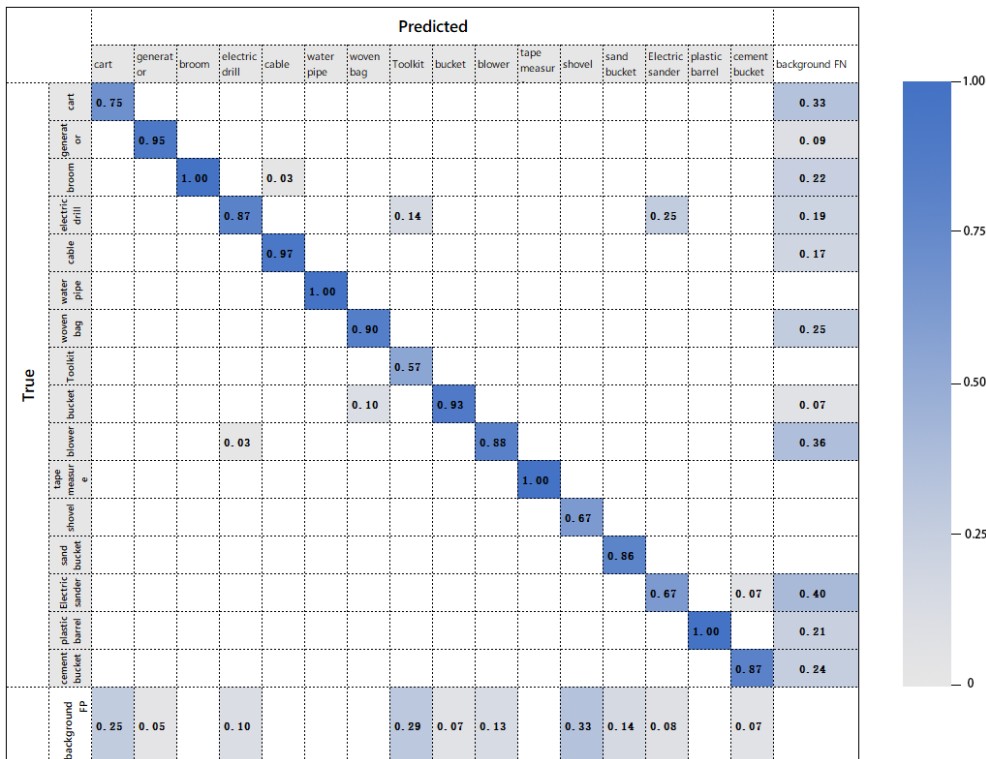

**Figure 7.** Confusion matrix for our model (columns are normalized).

### 4.3.1. Inspection Quality

In Figures 8–10, we compare some detection tool results of our model with Retinanet, YOLOv5s, and YOLOX in high-, medium-, and low-brightness image environments. Column a is the test result of Retinanet, column b is the test result of YOLOv5s, column c is the test result of YOLOX, and column d is the test result of our model. After comparison, we can see that in the first row of Figure 8, the detection results of our model are roughly the same as those of YOLOv5, and both are better than those using Retinanet and YOLOX. In terms of details, the detection grinder of our model has better adaptability. In the second row, our model has the best detection results, detecting 10 target objects with high accuracy. In Figure 9, for the detection results of medium-brightness images, compared to the first row, when the other three networks only detected 5 objects, our model has a higher detection rate in the face of wires with complicated overlapping situations. Comparing the second line, it can be seen that although Retinanet has completed the detection of all objects, the detection of woven bags has the same false detection as YOLOv5s, and YOLOv5s detects one fewer wire, while YOLOX detects two fewer wires; meanwhile, our model detected all items with high detection accuracy. In Figure 10. we compared the detection capabilities of the four networks in the dark environment. In the dark environment, our model has enhanced brightness. Compared with the first line, Retinanet has the fewest detection results, while for YOLOv5s, although there are many detection results, there are many false detections for shovels and wires. Compared with YOLOX, our model has better robustness in the face of highly overlapping wires, and the detection rate is higher than that of YOLOX. Comparing the second row, it can be clearly seen that the detection results of our model and YOLOv5s are still relatively complete, Retinanet lacks the detection of the blower, and YOLOX fails to detect a cement bucket in the dark. On the whole, our model has a higher detection ability than Retinanet, YOLOv5s and YOLOX in the full time domain and multi-superposition environment.

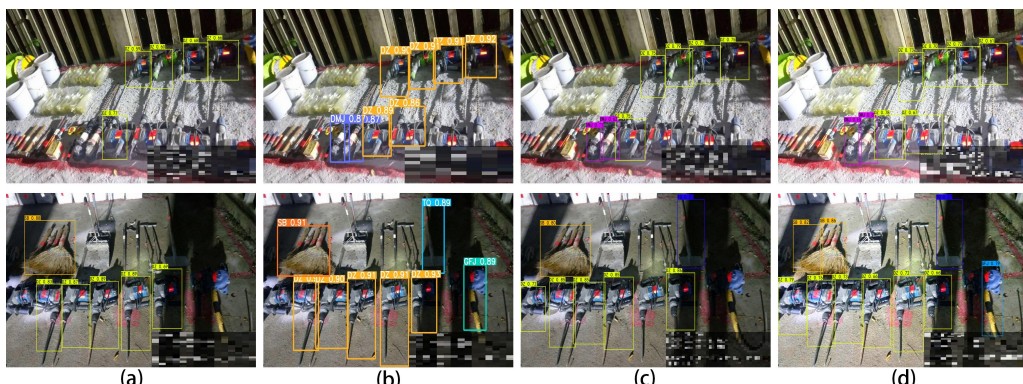

**Figure 8.** Comparison chart of high-brightness image detection results. (**a**) Retinanet. (**b**) YOLOv5s. (**c**) YOLOX. (**d**) Our model.

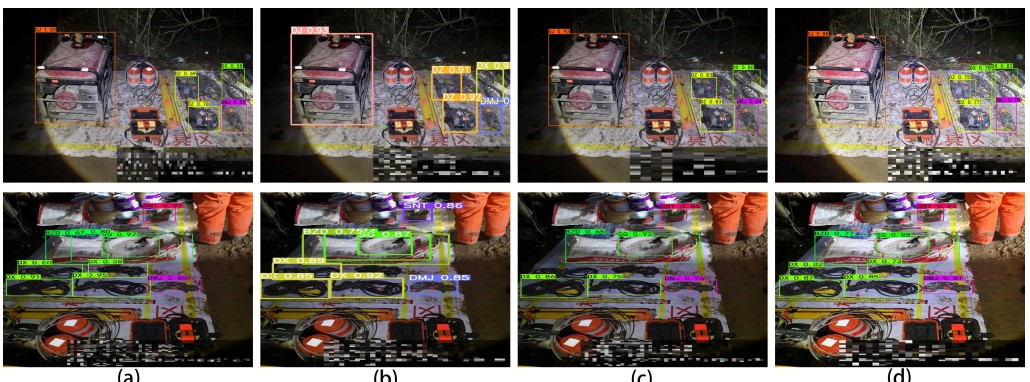

**Figure 9.** Comparison chart of medium-brightness image detection results. (**a**) Retinanet. (**b**) YOLOv5s. (**c**) YOLOX. (**d**) Our model.

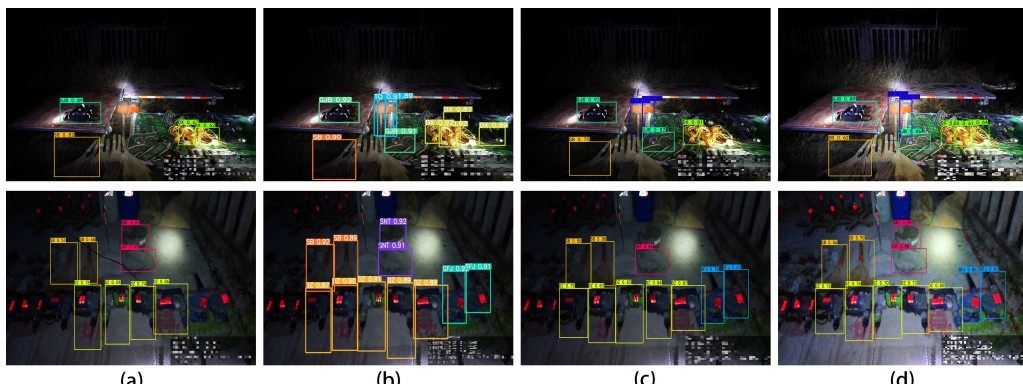

**Figure 10.** Comparison chart of low-brightness image detection results. (**a**) Retinanet. (**b**) YOLOv5s. (**c**) YOLOX. (**d**) Our model.

### 4.3.2. Ablation Experiments

To analyse the improved model performance of this paper, three groups of experiments have been designed to analyse the different improvements. Each group of experiments is tested on the same training parameters and different model contents. The detection results of the model performance are shown in Table 1. Compared with the first and second lines in Table 1, after the original YOLOX has the image enhancement module added, the model's detection ability of full time domain images is improved, and mAP improves by 0.97%. By comparing the experimental results of the second and third lines, after the loss function is modified to CIoU, mAP improves by 1.81% again. Further comparing the experimental results of the third and fourth lines, adding the lightweight attention module can improve the communication ability between channels, while weakening the noise impact on the

deep network, and the mAP improves by 0.32%. This shows the validity and rationality of the improved model in this paper.

**Table 1.** Experimental results of different improvement methods.

| Methods | mAP | FPS |
|---|---|---|
| Yolox | 74.16% | 34.26 |
| Yolox + Image enhancement | 75.13% | 35.58 |
| Yolox + Image enhancement + CIoU | 76.94% | 34.58 |
| Yolox + Image enhancement + CIoU + Attention module | 77.26% | 32.25 |

4.3.3. Comparison Experiment of Model Performance

To verify the detection performance of our model, we compared it with mainstream target detection models such as Retinanet, YOLOv5 and YOLOX. Among them, our models yolox and Retinanet adopt a confidence level of 0.001 and an iou of 0.65. YOLOv5 adopts official parameters (confidence = 0.25, iou = 0.45). The comparison results of various tools are shown in Table 2. According to the analysis in Table 2, the mAP of our model reaches 77.26%, which is 3.1% higher than that of the original YOLOX. Based on the analysis of the average AP values of the plastic bucket, motor, electric drill, grinding machine and woven bag in the table, it can be seen that the AP values of some tools detected by our method are improved to varying degrees compared with the original YOLOX model and have better detection performance compared with other mainstream target detection models (Retinanet and YOLOv5s). In particular, our model has obvious advantages in toolkits with many overlaps. Meanwhile, while ensuring high-precision detection, the FPS of the model does not decrease significantly, and the detection speed still has certain advantages compared with mainstream models.

**Table 2.** Experimental results of different improvement methods.

| Model | AP Plastic Bucket | Motor | Electric Drill | Grinding Machine | Woven Bag | Toolkit | mAP | FPS |
|---|---|---|---|---|---|---|---|---|
| Retinanet | 0.97 | 0.89 | 0.67 | 0.14 | 0.55 | 0.80 | 71.02% | 23.67 |
| Yolov5s | 0.65 | 0.91 | 0.34 | 0.04 | 0.53 | 0.99 | 68.37% | 41.37 |
| Yolox | 0.88 | 0.91 | 0.66 | 0.53 | 0.15 | 0.71 | 74.16% | 34.26 |
| Ours | 0.99 | 0.92 | 0.75 | 0.61 | 0.57 | 0.84 | 77.26% | 32.25 |

**5. Summary**

For the efficient detection of construction tools in complex railway work operation scenarios, this paper uses the images of tools and tools collected in actual railway operation and maintenance scenarios as the data set. The brightness enhancement network is improved to adapt to the operation and maintenance scene under complex lighting conditions and is merged with the object detection network. At the same time, we add a lightweight attention module to YOLOX and use the CIoU loss function to make the detection targeted and better distinguish overlapping tools. From the experimental results, the RYOLO proposed in this paper has better detection accuracy than YOLOX, and mAP increased by 3.16%. Furthermore, in the face of stacking tools, such as woven bags, electric drills, etc., it has a higher detection rate, which can meet the requirements of tool detection in complex environments on railways. At the same time, it realizes a higher intelligence of the railway operation and maintenance environment, reduces the hidden danger caused by the loss of tools and equipment, and ensures the safe operation of the railway. In the next step, to continue improving the detection ability, it is necessary to optimize further the railway tool data set and refine the full-time domain classification model.

**Author Contributions:** Investigation, Z.Z. and X.L.; Methodology, Z.Z. and Y.L.; Resources, S.L. and J.J.; Software, Y.L., S.L., Z.F., M.L. and Z.W.; Validation, Z.Z., S.L., Z.F. and J.J.; Visualization, Y.L. and X.L.; Writing—original draft, Y.L., M.L. and Z.W.; Writing—review & editing, Z.Z. and Z.F. All authors have read and agreed to the published version of the manuscript.

**Funding:** This work was funded by the Science and Technology Research Project of Jiangxi (China) Provincial Department of Education (No.GJJ202516).

**Institutional Review Board Statement:** Not applicable.

**Informed Consent Statement:** Not applicable.

**Data Availability Statement:** The source code and data: https://github.com/Riiiseee/Objects-Rapid-Detection-of-Railway-Works-in-Full-Time-Domain (accessed on 27 September 2022).

**Conflicts of Interest:** The authors declare no conflict of interest.

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
