# Peer review of "Rapid Detection of Tools of Railway Works in the Full Time Domain"

_sustainability, doi:10.3390/su142013662_

Round 1
Reviewer 1 Report
1. The literature review is too limited. The references are just cited in a row, without explaining the main features of each reference. More references should be added and the literature review should point out what has already done and which are the open issues and why they are so important, At the moment there is a list of papers.
2. Urge and challenges of the current situation have been discussed in the introduction, in addition, it is necessary to include Why the existing studies failed to explore these topics and how this study has an advantage over those studies in dealing with concern topic.
3. There are many things that is not clearly explain in this paper. The result and discussion, also the conclusion is short, and not answer the research questions (which also not clearly stated).
Reviewer 2 Report
Brief summary
This paper addresses an important issue, the use of deep learning in construction tool detection. The article is well written and addresses a wide audience, but needs some improvement:
Broad comments
Please, rewrite the abstract to make it clearer and more specific.
Please develop the introduction further, give a general background of what has been done so far, as it is difficult to determine what the article adds to science. Also show and highlight the research gap. Add more references to other studies on this issue. Maybe a deeper literature review would help?
Also work on the discussion, show what results other researchers have achieved and what you have achieved and why they differ.
Describe what software you used, what programming libraries you used, etc.
Please highlight in conclusions your results and be more specific.
Specific comments
Figure 7. Please correct figure to be more readable.
Also improve the graphic design of the other figures and their descriptions so that you do not have to look at the text to understand them.
